# Primary Structure and Coding Genes of Two Pheromones from the Antarctic Psychrophilic Ciliate, *Euplotes focardii*

**DOI:** 10.3390/microorganisms10061089

**Published:** 2022-05-25

**Authors:** Claudio Alimenti, Annalisa Candelori, Yaohan Jiang, Pierangelo Luporini, Adriana Vallesi

**Affiliations:** Laboratory of Eukaryotic Microbiology and Animal Biology, School of Biosciences and Veterinary Medicine, University of Camerino, 62032 Camerino, MC, Italy; claudio.alimenti@unicam.it (C.A.); annalisa.candelori@unicam.it (A.C.); yaohan.jiang@unicam.it (Y.J.); piero.luporini@unicam.it (P.L.)

**Keywords:** protein pheromones, macronuclear gene structures, polar microbiology, water-borne chemical signals

## Abstract

In ciliates, diffusible cell type-specific pheromones regulate cell growth and mating phenomena acting competitively in both autocrine and heterologous fashion. In *Euplotes* species, these signaling molecules are represented by species-specific families of structurally homologous small, disulfide-rich proteins, each specified by one of a series of multiple alleles that are inherited without relationships of dominance at the *mat*-genetic locus of the germinal micronuclear genome, and expressed as individual gene-sized molecules in the somatic macronuclear genome. Here we report the 85-amino acid sequences and the full-length macronuclear nucleotide coding sequences of two pheromones, designated E*f*-1 and E*f*-2, isolated from the supernatant of a wild-type strain of a psychrophilic species of *Euplotes*, *E. focardii*, endemic to Antarctic coastal waters. An overall comparison of the determined *E. focardii* pheromone and pheromone-gene structures with their homologs from congeneric species provides an initial picture of how an evolutionary increase in the complexity of these structures accompanies *Euplotes* speciation.

## 1. Introduction

Like many other organisms, from bacteria [1,2,3,4] to animals [5,6], ciliates synthesize cell-type specific signaling pheromones in functional association with their binary or multiple ‘mating type systems’, which control the cell switching from the vegetative (growth) stage to the sexual (mating) stage of the life cycle [7,8], and thus play a central role in speciation by driving the interpopulation gene flow [9,10]. Pheromonal activity has been identified in the cell-culture supernatant of various ciliates, either on the basis of mating induction assays in species of *Blepharisma*, *Dileptus*, *Euplotes* and *Oxytricha* ([11,12,13,14] as reviews), or observations of context-specific changes in cell morphology and behavior in species of *Ephelota* [15] and *Tokophrya* [16]. However, a successful pheromone isolation and chemical characterization has been carried out in only two cell types (I and II) of *Blepharisma japonicum* [11], and multiple cell types of five species of *Euplotes*, first of *E. raikovi* and then, *E. octocarinatus*, *E. nobilii*, *E. petzi* and *E. crassus* [12,13,14].

The two *B. japonicum* pheromones (originally designated ‘gamone 1’ or blepharmone’, and ‘gamone 2’ or ‘blepharismone’) are uniquely to be represented by chemically unrelated molecules, one being a glycoprotein of 272 amino acids and seven sugars, and the second a tryptophan derivative that has been reported to be synthesized also by other species of *Blepharisma* in addition to *B. japonicum* [17,18]. On the other hand, *Euplotes* pheromones have all been revealed to be acidic, cysteine-rich proteins varying in sequence length from 37 amino acids in *E. petzi* [19] to 105 in *E. octocarinatus* [20]. In *E. octocarinatus* and *E. crassus,* from which pheromones are purified in minimal amounts (0.4–0.5 µg of protein/liter of culture supernatant in *E. octocarinatus*), the pheromone structural knowledge has remained limited to the primary amino acid sequences [20,21,22,23]. The three-dimensional conformations have instead been determined, by NMR spectroscopy and/or X-ray crystallography, for seven *E. raikovi* pheromones (E*r*-1, E*r*-2, E*r*-10, E*r*-11, E*r*-13, E*r*-22 and E*r*-23), four (E*n*-1, E*n*-2, E*n*-6 and E*n*-1) of *E. nobilii*, and one (E*p*-1) of *E. petzi*, that are overall purified in larger amounts (up to 200–300 µg of protein/liter of culture supernatant in *E. raikovi*) [19,24,25,26,27,28,29,30]. Sharing a common helical fold tightly stabilized by densely spaced and tightly conserved disulfide bonds while differing from one another in local structural specificities, these conformations provided definitive evidence of the close relationships of structural homology that make *Euplotes* pheromones representative of species-specific protein families, and capable of competing with one another to bind target cells in both autocrine (self) fashion to promote cell growth and heterologous (non-self) fashion to induce cell mating [30,31]. This structural homology is the result of a genetic determination operated by multiple series of alleles that are inherited in Mendelian fashion at the so-called *mat* (mating-type) genetic locus of the chromosomic micronuclear (germinal) genome [32,33,34]. These allelic genes are then expressed with no mutual relationship of dominance in the sub-chromosomic macronuclear (somatic) genome [23], where they take the form of linear gene-sized DNA molecules including a single central coding region flanked by 5′-leader and 3′-trailer regions capped with telomeric C_4_A_4_ repeats [35,36].

Through a combined biochemical and genetic approach, we have determined and report the amino acid sequences and the structures of the macronuclear coding genes of two pheromones that are secreted in tiny amounts by a rather uncommon *Euplotes* species, *E. focardii*, which is characterized by a strictly psychrophilic behavior being endemic to the Antarctic coastal waters [37], and a position in the *Euplotes* phylogenetic tree distant from the basal clades including *E. petzi*, *E. raikovi* and *E. nobilii*, and close to the late clades including *E. octocarinatus* and *E. crassus* [38,39]. An overall comparison of the pheromone and pheromone–gene structures between *E. focardii* and the other *Euplotes* species suggests an initial picture of how the evolution of these structures accompanies *Euplotes* speciation.

## 2. Materials and Methods

### 2.1. Pheromone Source

The pheromone source was the *E. focardii* strain TN_1_, which was cultivated on the green alga *Dunaliella tertiolecta*, at 2–4 °C, under a rhythm of 16 h of darkness and 8 h of very weak light. This strain (deposited at the Culture Collection of Algae and Protozoa, CCAP, Scottish Marine Institute, Argil PA37 1QA, Scotland, under the accession number 1624/17) was established in the laboratory starting from one of several specimens isolated from a sample of sandy sediments collected in January 1988 from a small cove close to the Italian research station at Terra Nova Bay (Ross Sea, Antarctica) [37]. Supernatant preparations for the pheromone isolation were obtained from cultures that were expanded by continuous additions of moderate concentrations of food, to reach a final volume of 15–20 L and a cell density of approximately 2 × 10^3^ cells/mL, and were then left to starve for 1 week. Supernatant was prepared by removing cells and debris in suspension through filtration.

### 2.2. Pheromone Purification and Analysis 

The pheromone purification was carried out according to a standard three-step chromatographic procedure, previously set up for the pheromone purification from other marine species of *Euplotes* [22,40]. Proteins were adsorbed from supernatant preparations onto reverse-phase Sep-Pak C_18_ cartridges (Waters, Milford, CA, USA) and separated, first, by gel filtration on a Superdex peptide HR10/30 column (GE Healthcare, Little Chalfont, UK), equilibrated in Tris-HCl 20 mM and NaCl 0.4 M and eluted at 0.4 mL/min; then, by reverse-phase-high-performance chromatography [RP-HPLC] on a 4.6-mm × 250-mm C_18_ column (Supelco, Bellefonte, PA, USA). Elution was carried out with a discontinuous acetonitrile gradient in the presence of 0.06% (*v/v*) trifluoroacetic acid. The molecular mass measurements were performed by matrix-assisted laser desorption/ionization [MALDI] time-of-flight [TOF] mass spectrometry, using a Voyager DE-PRO spectrometer (Perspective Biosystems, Framingham, MA, USA), equipped with a nitrogen laser operated at 337 nm. The protein automated Edman degradation was carried out in a Procise 492 protein sequencer (Applied Biosystem, Perkin Elmer, Foster City, CA, USA).

### 2.3. Assaying Pheromone Activity

Ciliate pheromones are conventionally detected in solution via mating induction assays, which entail the capacity of genetically identical cells to form homotypic (or selfing) mating pairs in response to the suspension with heterologous pheromone solutions [11,13]. Among *Euplotes* species, this capacity is, however, native to only some species and alien to others including *E. focardii* which, in addition, is rather impracticable for mating studies due to the enormous lengthening that its life in freezing waters imposes on the dynamics of the mating process [41]. The assays for the identification of the chromatographic peaks containing pheromone activity were thus carried out, in a heterospecific manner, on *E. raikovi* cells, which are constitutively committed to practice homotypic pairing and were previously used to identify pheromones from another species, *E. crassus*, which, like *E. focardii,* is refractory to mate in homotypic fashion [22]. Samples of 2 × 10^3^ cells from *E. raikovi* type-I cultures, previously deprived of food for a couple of days, were resuspended with 500-µL of fresh seawater and incubated with three increasing volumes (10, 20 and 40-µL) from the chromatographic fractions of interest. The mixtures were then inspected over an interval of 1–2 days to detect the cell mating response. This response was also taken as positive in the case of the formation of unstable cell–cell unions in mating pairs.

### 2.4. Gene Amplification, Cloning and Sequencing

Gene amplification was carried out by applying a Rapid Amplification of Telomeric Ends [RATE]-Polymerase Chain Reaction [PCR] protocol to macronuclear DNA preparations obtained from cell pellets following a standard procedure [42]. PCR primer designations and sequences are listed in Table 1. The reactions were run in an Eppendorf Ep-gradient Mastercycler (Eppendorf AG, Hamburg, Germany), using 0.5 μg DNA aliquots in 50 μL-reaction mixtures containing 0.5 μM of each primer, 0.25 mM dNTP and 0.02 U/μL of Phusion High-Fidelity DNA Polymerase (Thermo Fisher Scientific Inc., Waltham, MA, USA). As a rule, 35 amplification cycles were carried out. Each cycle consisted of a 98 °C denaturation step for 30 s, a 30 s annealing step, and a 72 °C elongation step for 30–90 s. The temperature of the annealing step varied from 55 to 65 °C, depending on the primer G + C content. A final incubation step, at 72 °C for 5 min, was added to the last cycle. Amplified products were cloned in the pJET1.2/blunt cloning vector (Thermo Fisher Scientific Inc., Waltham, MA, USA), following the procedure suggested by the manufacturer.

## 3. Results

### 3.1. Pheromone Isolation and Structural Characterization

The protein material extracted from the supernatant preparations of the *E. focardii* cultures and initially separated on gel filtration revealed pheromonal activity, as detected by heterospecific mating induction assays, in association with only four fractions in the range from 9 to 12 kDa (Figure 1a). On reverse-phase chromatography, the protein pool of these fractions was next resolved into two major peaks (Figure 1b), each containing a single protein species as assessed by mass spectrometry analysis (Figure 1c,d). One species was measured to be 9245.5 Da, and was identified with one pheromone designated E*f*-1; the second species was measured to be 9201.7 Da, and identified with a second pheromone designated E*f*-2. The two proteins were quantified in the cell-culture supernatant at a concentration of 10–20 µg/L, and their co-existence provided direct evidence that their source TN_1_ cells carried a heterozygous allelic combination of the *mat* locus of the micronuclear genome, as in general is the case in wild-type *Euplotes* strains [9,34].

The determination of the E*f*-1 and E*f*-2 amino acid sequences involved a combined chemical and genetic approach. By subjecting aliquots of the two purified proteins to automated Edman degradation, a N-terminal 15-amino acid sequence, Ser_1_-Asp-Cys-His-Gly-Asp-Thr-Glu-Tyr-Leu-Ile-Asp-Asp-Glu-Ser_15_, was preliminarily obtained. This sequence was identical for the two proteins and included one Cys residue indirectly identified as ‘no signal’. The complete E*f*-1 and E*f*-2 amino acid sequences were then deduced from the nucleotide sequences of the respective macronuclear coding genes, designated *mac-ef-1* and *mac-ef-2*, which were amplified and cloned via a nested RATE-PCR procedure.

The 3′-trailer region and a 218-bp portion of the coding region of the *mac-ef-1* and *mac-ef-2* genes were obtained by sequencing amplicons of a similar 400-bp size generated by a first round of DNA amplifications, in which two degenerate oligonucleotides (“dFW1” and “dFW2”, Table 1) designed on the determined N-terminal amino acid sequence were used as forward primers in combination with one oligonucleotide (“TEL”, Table 1) specific to the telomeric repeats distinctive of the 5′ and 3′ telomeric ends of *Euplotes* macronuclear genes [35,36].

The 5′-leader region and the remaining portion of the coding region of the two pheromone genes were next obtained by sequencing 1650-bp and 850-bp amplicons generated from a second round of DNA amplifications run with oligonucleotides (“RV1” and “RV2”, Table 1) designed on the 400-bp amplicon sequence and used as reverse primers in combination with the TEL oligonucleotide. The 1650-bp amplicon sequence, terminating with 5′ telomeric repeats, allowed a direct reconstruction of a full-length 1882-bp gene sequence (telomeres excluded), which was recognized to be specific of pheromone E*f*-2 in relation to its open reading frame (ORF). It encoded a 120-amino acid pre-pro-pheromone precursor in which the calculated molecular mass of the secreted 85-amino acid protein closely matched the 9201.7 Da mass previously measured for the isolated E*f*-2 protein. 

At the opposite of the 1650-bp amplicon, the 850-bp amplicon lacked 5′-end telomeric repeats, and efforts to amplify the whole 5′-leader region necessary to reconstruct the second full-length pheromone gene sequence were unsuccessful. To identify this sequence, the *E. focardii* genome assembled from the same cells of the TN_1_ strain used in this work [43] was helpful. Queried on this genome, the 850-bp amplicon sequence localized within a 2318-bp gene sequence (MJUV02005382.1), which was thus recognized to be specific of pheromone E*f*-1. Its ORF encoded a 126-amino acid pre-pro-pheromone precursor in which the calculated molecular mass of the secreted 85-amino acid protein closely matched the 9245.5 Da mass previously measured for the isolated E*f*-1 protein. The *mac-ef-1* gene sequence was then validated by sequencing the 2253-bp fragments obtained by DNA amplifications run with two oligonucleotides (“5′-FW1” and “3′-RV” in Table 1) designed near the 5′ and 3′ telomeric ends of the identified genomic sequence (Figure 2b). 

Instead, the query of the 1882-bp *mac-ef-2* sequence on the *E. focardii* genome identified only a partial 1355-bp sequence (MJUV02014221.1) equivalent to the gene 5′-leader region. The gene validation was obtained by sequencing 1778-bp fragments from a DNA amplification run with two oligonucleotides (“5′FW2” and “3′RV” in Table 1) designed close to the gene 5′ and 3′ telomeric ends (Figure 2b).

### 3.2. Pheromone Gene Structure

The ORFs spanning from A_1769_TG to TAA_2149_ in the *mac-ef*-1 gene, and from A_1358_TG to TAA_1720_ in the *mac-ef*-2 gene, were unequivocally identified as specific for the synthesis of cytoplasmic (immature) pheromone precursors, pre-pro-E*f*-1 and pre-pro-E*f*-2 of 126 and 120 amino acids, respectively, destined to be proteolytically processed to remove the signal-peptide and pro-segment before the secretion of the mature pheromone forms (Figure 2c). The coding sequences of these pheromone precursors vary only for a deletion of an 18-bp segment responsible for a shorter pro-segment in the *mac-ef-*2 gene (16 amino acids vs. 22 in the pro-segment of pre-pro-E*f*-1), and eight nucleotide mutations that are responsible for seven amino acid substitutions. One localizes in the 19-amino acid pre-segment, three in the 16- or 22-amino acid pro-segment, and three in the 85-amino acid secreted protein. 

At the level of the two ORF flanking regions, the *mac-ef*-1 and *mac-ef-*2 genes are characterized by a tight conservation (96% sequence identity) of their relatively short 3′-trailer regions (169 and 162 bp, respectively), lacking a canonical AATAAA polyadenylation motif, which is likely replaced with a TTATTT motif located 15 bp downstream the ORF TAA stop codon. On the other hand, the 5′-leader regions are poorly conserved (42% sequence identity) and markedly extended (1768 bp in *mac-ef*-1, and 1357 bp in *mac-ef*-2). This poor conservation, that has no equal in allelic pheromone genes characterized from other *Euplotes* species [19,20,21,28,44], is also due to the loss in the *mac-ef-2* gene of five segments of 73, 90, 93, 146 and 169 bp from the 5′-leader region (Appendix A). The remarkable extension of this region likely reflects the inclusion of intron sequences, as suggested by the presence of multiple canonical GTA-TAG splicing sites.

### 3.3. Pheromone Primary Structure

The 85-amino acid E*f*-1 and E*f*-2 sequences, which result from the cytoplasmic precursors by proteolytic removals of the signal peptide and pro-segment at predicted Gly-Phe and Gly-Ser sites, respectively, are closely similar (Figure 3a). They differ for only three amino acid substitutions, Glu_27_/Ala, Thr_28_/Met and Tyr_52_/Phe, the first two of which credit more structural relevance, since the substitution of charged and polar residues with hydrophobic residues locally alters the molecule physico-chemical properties [45]. Both sequences start with a Ser residue, are strongly acidic with an isoelectric point of 3.8, and include ten unevenly spaced Cys residues, destined to pair into intra-chain disulfide bridges warranting a global tight stabilization of the E*f*-1 and E*f*-2 molecular structures. Four cluster into a CX_1-3_CCX_2-3_ C motif (X stands for any residue) that finds a precise counterpart in *E. crassus*, *E. nobilii* and *E. petzi* pheromones (Appendix A) [19,23,28].

In line with cold-adapted *E. nobilii* pheromones [28,46] and psychrophilic proteins in general [47], E*f*-1 and E*f*-2 amino acid composition strongly reflects a molecular adaptation to low temperatures, which in the first place requires weakening the packing of the molecular structure and an improved molecular backbone flexibility. Compared with the pheromones of the mesophilic *Euplotes* species, the very hydrophobic residues (Ala, Ile, Leu, Met, Phe, Pro, Val and Trp), which preferentially bury into the protein structural core, are under-represented (or not represented, Trp), overall accounting for only 24.7% of the amino acid composition. In contrast, charged and polar residues are largely preferred, in particular Asp, Thr, Tyr and Ser residues, which, together with 11 Gly residues (neutral), constitute exactly half the amino acid composition. Their overrepresentation, Asp residues (amounting to 12!) in particular, finds a close parallel in psychrophilic proteins of bacteria [47], and is justified by a key role in improving the backbone structural flexibility by avoiding inclusion into helices, or favoring their destabilization [48,49].

The hydropathy plot (Figure 3b) of the E*f*-1 and E*f*-2 amino acid sequences unequivocally identifies the amino-terminal and central regions as mostly characterized by three strongly hydrophilic and poorly structured districts. The major one spanning 20 residues from Cys-V to Cys-VI includes six of the 11 Gly residues, and each is associated with a local increase in solvent accessibility and molecular backbone flexibility (Figure 3c,d). The carboxy-terminal region is instead identified as the major hydrophobic district preceded by one (E*f*-1 sequence) or two (E*f*-2 sequence) minor hydrophobic sites, all coincident with a reduced solvent accessibility and a more rigid structure. The likely organization of these hydrophobic sites into helical secondary structures is strongly supported in particular for the major one. Its counterparts in the carboxy-terminal region of the determined *Euplotes* pheromone molecular structures are all shown to be organized into a regular α-helix, which is tightly conserved at both intra- and inter-specific levels, serving a key function in the pheromone/receptor interactions [30].

## 4. Discussion

The structural characterization of *E. focardii* pheromones and pheromone coding genes is an initial step to outlining a picture of how these molecular structures evolve in concert with *Euplotes* speciation. Using a simplified version of the *Euplotes* phylogenetic tree commonly proposed with seven distinct clades, annotated from I to VII starting from the earliest branching one [39], Figure 4 delineates the phylogenetic relationships and ecological diversification of *E. focardii* and the five other *Euplotes* species (*E. crassus*, *E. nobilii*, *E. octocarinatus*, *E. petzi* and *E. raikovi*) with known pheromone and pheromone coding gene structures [12,13,19]. It appears that *E. petzi* (marine, polar) and *E. crassus* (marine, temperate) represent the two more distantly related species, the former splitting into the basal clade I and the latter into the terminal clade VII. The four other species instead separate into in-between branching clades: *E. rakovi* (marine, temperate) and *E. nobilii* (marine, polar) into clade IV; *E. octocarinatus* (lacustrine, temperate) into clade V including only freshwater species; and *E. focardii* (marine, polar) into the widest and least resolved clade VI.

The overall interspecific comparison of the pheromone and pheromone gene structures (which in the figure are typified alongside each species) provides evidence for an evolutionary tendency of these structures to jointly increase in extension and complexity.

In the structural evolution of the pheromone genes, the most manifest variations coincide with the transition from *E. petzi* to the other species and primarily involve the 5′-leader region or, uniquely in *E. octocarinatus*, the coding region, while the 3′-trailer region is left practically untouched. The 5′-leader region of minimal length (72 bp) in *E. petzi*, at odd with much more extended coding and 3′-trailer regions (219 and 368 bp, respectively), abruptly lengthens from 5 to 7 folds in *E. raikovi*, *E. nobilii* and *E. crassus*—which is distinguished by a duplicated micronuclear *mat*-gene locus [23]—and up to 24 folds in *E. focardii*. The result is that it largely overcomes in length the coding and 3′-traler regions that remain substantially unvaried. Only in *E. octocarinatus*, the 5′-leader region eccentrically maintains a minimal length as in *E. petzi,* and its contribution to the gene sequence lengthening is replaced by the coding region that becomes 3 folds longer than in all the other species.

These increases in the extension of the pheromone genes are explained by the findings [21,44] that the 5′ leader region of the *E. raikovi* pheromone genes and the coding region of the *E. octocarinatus* pheromone genes include intron sequences (up to 514 bp long in *E. octocarinatus*). They thus assume strong functional relevance considering the central role that the splicing of these intron sequences plays in the mechanism of pheromone gene expression. Although intron sequences have not directly been identified in the 5′-leader region of the *E. nobilii*, *E. focardii* and *E. crassus* pheromone genes, their presence is strongly supported by the inclusion of multiple canonical GTA/TAG splicing motifs which, instead, are completely lacking throughout the *E. petzi* pheromone gene sequences [19].

Similar to the pheromone gene structural evolution being shown to primarily operate by widening the 5′-leader region or the coding region with the inclusion of intron sequences, the pheromone structural evolution appears to primarily operate by extending one or more sequence segments spanning preferentially between conserved Cys residues lying in the sequence middle (Appendix A). By including new amino acid combinations, these extensions clearly provide *Euplotes* pheromones with the necessary structural substrate to adapt and optimize their activity in any new environment.

No inter-cysteine segment includes more than four residues in *E. petzi* pheromones; a maximum of eight or nine residues are included in two inter-cysteine segments of *E. raikovi* pheromones, and systematically more than 10 (up to 25) residues are included in one or more inter-cysteine segments of pheromones of all the other species. These local sequence extensions have functionally been related to cold-adaptation in *E. focardii* on the basis of predictive algorithms applied to the determined E*f*-1 and E*f*-2 amino acid sequences. This association was previously more directly supported by comparing the pheromones of the two phylogenetically closely related marine species, the polar *E. nobilii* and the temperate *E. raikovi*, for their three-dimensional structures [46] and the dynamics of thermal unfolding and refolding [55]. Most likely, however, the functions of the local pheromone sequence extensions are multiple and varied, and their identification requires the determination of the pheromone three-dimensional structures from a wider spectrum of ecologically well-distinct *Euplotes* species.

## Figures and Tables

**Figure 1 microorganisms-10-01089-f001:**
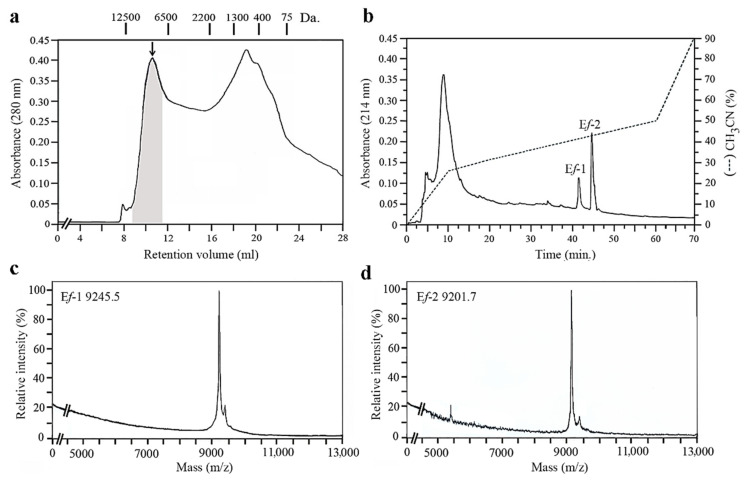
Purification and mass spectrometry of pheromones E*f*-1 and E*f*-2. (**a**) Elution profile of the gel filtration chromatography of the protein material adsorbed onto reverse-phase cartridges from cell-culture supernatant preparations. In the chromatogram, the shadowed area indicates the pheromone-containing fractions. The points of elution of proteins with standard molecular masses are indicated above the chromatogram. (**b**) Elution profile of the protein material pooled from the shadowed fractions and separated on reverse-phase chromatography. The applied acetonitrile gradient is indicated by a dotted line and the elution peaks of the two proteins identified as pheromones are labelled E*f*-1 and E*f*-2. (**c**) Molecular mass determination of pheromone E*f*-1. (**d**) Molecular mass determination of pheromone E*f*-2.

**Figure 2 microorganisms-10-01089-f002:**
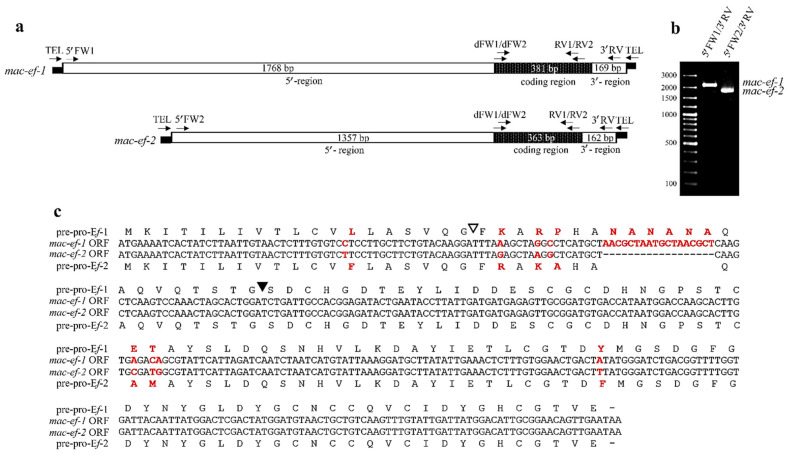
Structure of the *mac-ef-1* and *mac-ef-2* genes. (**a**) Schematic representations of the two genes, in which bars represent the telomeric ends consisting of C_4_A_4_ and G_4_T_4_ repetitions, boxes represent the 5′-leader, coding and 3′-trailer regions of different lengths, and arrows indicate the directions and positions along with the denominations of primers used in the PCR gene amplifications. (**b**) Agarose gel electrophoresis of PCR fragments obtained by amplifications of *mac-ef-1* and *mac-ef-2* genes, using the indicated primer combinations. (**c**) Alignment of the *mac-ef-1* and *mac-ef-2* ORF nucleotide sequences, and deduced amino acid sequences of pre-pro-E*f*-1 and pre-pro-E*f*-2. Nucleotide and amino acid variations are highlighted in red. In the amino acid sequences, the light and filled arrowheads indicate the putative cleavage sites between the signal-peptide and pro-segment, and between the pro-segment and secreted pheromone, respectively.

**Figure 3 microorganisms-10-01089-f003:**
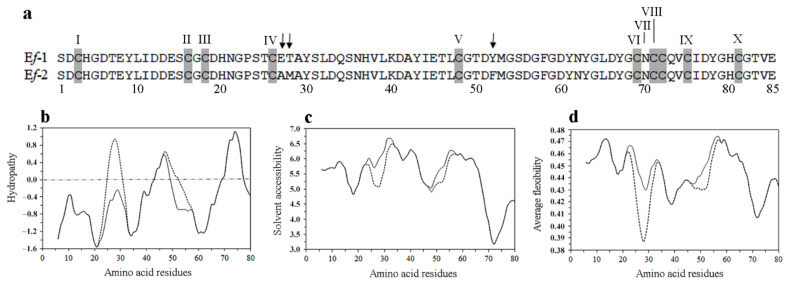
Amino acid sequence alignment and physico-chemical traits of E*f*-1 and E*f*-2 pheromones. (**a**) Sequence alignment with Cys residues, progressively numbered I to X, highlighted in grey and amino acid substitutions indicated by arrows. (**b**) Hydropathy, (**c**) solvent accessibility and (**d**) average flexibility plots of the E*f*-1 (solid line) and E*f*-2 (dotted line) pheromones, as generated by predictive algorithms [50,51,52].

**Figure 4 microorganisms-10-01089-f004:**
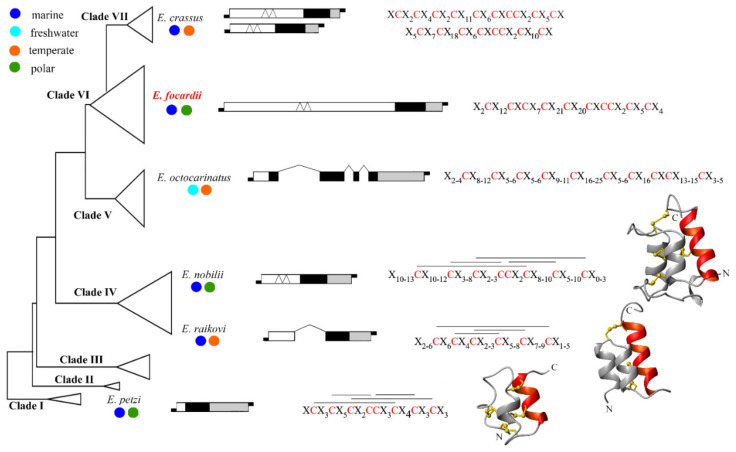
*Euplotes* pheromone and pheromone-gene structural evolution. The *Euplotes* phylogenetic tree, articulated on seven clades, is reported in a very simplified version to highlight the positions of the species with known pheromone and pheromone-gene structures. Indicated for each species is: (i) ecology, (ii) the pheromone gene structure, (iii) the pheromone amino acid consensus sequence, and (iv) the pheromone native three-dimensional structure, known from *E. petzi*, *E. raikovi* and *E. nobilii*. In the pheromone gene diagrams, drawn to scale except the telomeric extremities represented by bars, the 5′-leader, coding, and 3′-trailer regions are represented by empty, filled, and grey boxes, respectively. Inter-box zig-zag lines indicate intron-sequence splicing, as determined in the 5′-leader and coding regions of *E. raikovi* [44] and *E. octocarinatus* [21] pheromone genes, respectively. The likely presence of intron sequences (as deduced from intron-splicing motifs) in the 5′-leader region of the *E. nobilii*, *E. focardii* and *E. crassus* pheromone genes is indicated by intra-box zig-zag lines. In the amino acid consensus sequences, the conserved Cys residues are highlighted in red, X stands for any residue, and lines connect Cys residues according to their disulfide bridges. In *E. petzi*, the consensus sequence is pertinent to pheromones E*p*-1, E*p*-2, E*p*-3 and E*p*-4 [19]; in *E. raikovi*, to pheromones E*r*-1, E*r*-2, E*r*-4, E*r*-5, E*r*-6, E*r*-7, E*r*-8, E*r*-10, E*r*-11, E*r*-13, E*r*-20, E*r*-21, and E*r*-22 (the 51-amino acid eccentric family member E*r*-23 excluded) [26,44]; in *E. nobilii*, to pheromones E*n*-1, E*n*-2, E*n*-6, E*n*-A1, E*n*-A2, E*n*-A3, and E*n*-A4 [28]; in *E. octocarinatus*, to pheromones Phr1, Phr1*, Phr2, Phr2*, Phr3, Phr3*, Phr^−5^, and Phr^−2^ (the Phr4-amino acid eccentric family member excluded) [20,21,53]; in *E. focardii*, to pheromones E*f*-1 and E*f*-2 [this work]; in *E. crassus*, to pheromones E*c*-1, E*c*-2 and *Ec*-3 representing one sub-family, and to pheromone E*c*-α representing a second sub-family [23]. The pheromone three dimensional configurations, shown in frontal view and ribbon diagrams, are specific of pheromones E*p*-1 [19], E*r*-1 [24] and E*n*-6 [54] taken as representative of *E. petzi*, *E. raikovi* and *E. nobilii* pheromone families, respectively. The disulfide bridges are in yellow, the structurally more conserved helix 3 is highlighted in red, and N and C indicate the molecule amino- and carboxyl terminus, respectively.

**Table 1 microorganisms-10-01089-t001:** PCR primer designation and sequences.

Name	Nucleotide Sequence (5′–3′)
dFW1 ^a^	GATTGCCAYGGWGATACNGA
dFW2 ^a^	GATACTGAATAYTTMATYGAYGART
TEL	CCCCAAAACCCCAAAACCCC
RV1	TCAATACAAACTTGACAGCAGTTACA
RV2	TAATCACCAAAACCGTCAGATCCCAT
5′-FW1	TGGCTAAAGCATAGAATTCTAACAT
5′-FW2	TGTTTATAAATGAGGAAGTGCTTAAG
3′-RV	ACACTATTGAACCAGAATATTCCTCT

^a^ Y, W, M and R, alternatives between C and T, A and T, A and C, and A and G, respectively. N, any nucleotide.

## Data Availability

Macronuclear pheromone gene sequences are deposited in GenBank database under the accession numbers: *mac-ef-1*, ON428185; *mac-ef-2*, ON428186.

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
