# Peer review of "Primary Structure and Coding Genes of Two Pheromones from the Antarctic Psychrophilic Ciliate, Euplotes focardii"

_microorganisms, 2022, doi:10.3390/microorganisms10061089_

Round 1

Reviewer 1 Report

C. Alimenti et al. designed and executed a search for sexual mating inducing pheromone genes in Euplotes focardii. The tried-and-true approach of fractionating filtered supernatant followed by pheromone activity assay was well thought-out and successfully identified two pheromone proteins and their respective genes. As the authors point out, this discovery broadens the spectrum of identified pheromone genes within Euplotes, especially in a less defined Clade and provide further insights into the evolutionary path of mating signaling in Euplotes.

General comments:

All of the major results in the article are based on comparing the two identified pheromones and their genes to their homologs from other Euplotes species. However, in several cases, such comparisons are not clearly shown or not explained in a manner accessible to readers with limited understanding of the field.

Specific comments:

The current introduction is not sufficient for any readers who are not familiar with the decades of prior work on Euplotes pheromones to understand the significance of the results in the current manuscript. The introduction would therefore benefit from a significant expansion. Among other aspects, I would keenly suggest providing background including: a. an overview of Euplotes phylogeny and the position of E. focardii; b. features and motifs of Euplotespheromones, particularly the conserved Cys residues, and how they contribute to the known functions; c. if possible, current understanding of broad questions discussed in this work e.g. cold adaptations and other evolutionary changes in Euplotes genes.

Line 93: The results of the pheromone activity assay are only shown as the shadow area in Fig. 1a. If possible, the analysis of three volumes from the chromatographic fractions should be included in the supporting data, especially for the lower MW peak, to address whether other minor peaks may contain additional pheromone activity.

Line 184: The suggestion here is that the 850 bp amplicon corresponds to part of the 5’ region and most of the coding region of Ef-1 ending at RV1/RV2. This should be confirmed. With the genome assembly available, it should be attainable to confirm the full sequence of this gene by overlapping sequences of 2-3 PCR fragments (rather than a whole as in line 179).

The authors should also state whether the expected ef-2 gene was present in the assembly. These additions would give more confidence in the authors’ conclusions drawn from examination of the gene structures.

Line 234: The nature of the suggested intragenic recombination is unclear, especially when S1 is not accessible to this reviewer. It should be explained or illustrated in a figure.

Line 239: “the inclusion of intron sequences” hypothesis seems as plausible as the recombination hypothesis. Consider removing “Instead, ” to avoid confusion.

These different mechanistic options would be clearer if the authors showed a comparison between ef-1, ef-2 and the corresponding schematics from the referenced E. octocarinatus and E. raikovi genes (similar to those in Fig. 4, with added detail such as more granular alignment, predicted introns and so on) and perhaps all other five examples shown in Fig 4. Providing these comparisons more clearly would give better support to the conclusions regarding cold adaptation, gene structure evolution and so on.

Line 247: The conservation of the 10 Cys residues need to be more clearly demonstrated, e.g., in a more comprehensive alignment than in Fig. 3a. The same is true for the CNCCQVC motif in line 251. Showing these data might also help to support the discussion about the lengthening of the segments between Cys residues (line 360).

Line 252: It’s unclear how the change in hydropathy by the three aa differences are relevant structurally or functionally.

Line 276: The richness of polar and charged residues in the N-term region is reported to be relatively high. However, it’s not clear what sequences are being compared to draw this conclusion: N-term regions of pheromone gene products or general amino acid composition in Euplotes? Can the authors discuss how this feature in the N-term region would confer cold-adaptation?

Line 286: the comparison with the pheromones of the other three species is not shown.

Line 298: Based on the references cited, “Clade II” was added in 2021 thanks to a phylogenetic analysis of rRNA classifying the newly isolated species E. warren “into its own early branching clade” [42]. Describing the current understanding as “commonly viewed to be articulated on seven major clades numbered I to VII” seems like an over-simplification under the circumstances, so should be stated in a more balanced way. As suggested above, this might usefully be included the Background material to the manuscript. Importantly, the relative phylogenetic positions of the six species chosen are fairly supported in [41] and [42].

Line 305: “rakovi” should be “raikovi”

Line 365: I am confused by the authors’ observation that E. focardii and E. octocarinatus both form proposed unstructured domains that reflect “adaptive specificities”, considering that the two species are reported to live in dramatically different environments. Please clarify what is meant here.

Line 374: This statement about “rule” vs “exception” is confusing. There is no evidence presented to support 5’ region extension as a rule while coding region extension as an exception. Because of the ambiguity in the writing, it isn’t clear whether the authors infer that the 3-fold extension in other Euplotes was an intermediate that led to the 20-fold extension in E. focardii. These points may be better illustrated in a comparison of the schematic of these genes as suggested for line 239 above.

Author Response

We would sincerely thank the two Reviewers for the constructive criticisms and suggestions, which prompted us to substantially expand the Introduction, rewrite large parts of the Results and Discussion, improve the presentation of the iconography, and add supplementary material. In particular:
In relation to the major points raised by Reviewer #1.
1. As specifically required, the Introduction has been substantially widened to provide more background information on the structure and activity of ciliate, Euplotes in particular, pheromones.
2. In relation of the pheromone activity in the chromatographic fractions, the only four fractions containing active proteins have been specified and indicated in the Fig. 1a by a shadowed area.
3. The experimentally more relevant point was raised as following: “The suggestion here is that the 850 bp amplicon corresponds to part of the 5’ region and most of the coding region of Ef-1 ending at RV1/RV2. This should be confirmed. With the genome assembly available, it should be attainable to confirm the full sequence of this gene by overlapping sequences The authors should also state whether the expected ef-2 gene was present in the assembly. These additions would give more confidence in the authors’ conclusions drawn from examination of the gene structures.”. In the revised version, we have more exhaustively described the experiments carried out (PCR amplifications and sequencing) to confirm the full-length pheromone gene sequences, and added a new figure (Figure 2b) to document the results. Some speculative conclusions have been removed, and a new gene sequence alignment has been added as Supplementary Information (too long to be inserted in the main text) to highlight nucleotide identities, putative splicing sites, and a poly(A) signal.
4. In the Results, a new paragraph has been dedicated to the specificities of the Ef-1 and Ef-2 amino acid sequences, and the Discussion has in practice been completely rewritten to better rationalize the likely functional significance of the inter-species evolutionary variations in the pheromone and pheromone-gene structures. Lastly, as specifically requested, a new Supporting Figure S2 has been added showing the amino acid sequence alignment of E. focardii pheromones with E. nobilii and E. petzi pheromones carrying a tightly conserved Cys motif.
In relation to points raised by Reviewer #2.
1. A comparison of the pheromone activity (as measured in “arbitrary units” according to a conventional protocol originally set on Blepharisma pheromones/”gamones”) between E. focardii and other congeneric species is hardly practicable. The main reason of this impracticability is that this activity could be revealed only through heterospecific assays, using E. raikovi cells with a previously known capacity to unite in homotypic/selfing pairs in response to suspension with a non-self pheromone. In addition, any observation on mating in E. focardii would have required cultures of strongly mating compatible strains (no longer available), and hours in elaborating a
mating response (not days as is the case in E. focardii, adapted to freezing marine waters).
1. With regard to the suggestion on the E. focardii pheromone-gene cold-adaptation, comparative studies have mainly been focused on proteins and lipids from extremophile organisms and those related to genomes have been directed on identification of cold tolerance-associated genes in bacteria and vertebrates. An analysis of the cold-adaptation of E. focardii genes would acquire significance only in a wider context of genome cold-adaptation, which is the object of current research from other laboratories (see the quoted reference Mozzicafreddo et al. The macronuclear genome of the Antarctic psychrophilic marine ciliate Euplotes focardii reveals new insights on molecular cold adaptation. Sci Rep 2021, 11, 18782.)
2. All common molecular modeling software are, to varied extents, ineffective to picture a reliable model of the folding of Ef-1 and Ef-2 amino acid sequences, and fail also with other pheromone sequences, such as those of E. crassus and E. octocarinatus, including relatively long inter-cysteine sequences (that are cause of strong polymerization effects). When applied (as a sort of control) to E. raikovi, or E. nobilii pheromones with known three-dimensional conformations (as determined by NMR and X-ray analyses on native protein preparations), most software predict folding also into beta-sheets, whereas these pheromones fold exclusively in alpha helices.
3. As anticipated above, the Discussion has been consistently widened and largely re-written.

Reviewer 2 Report

The manuscript submitted by Alimenti et al. presented a study on the isolation and characterization of two pheromone genes from an Antarctic psychrophilic ciliate, Euplotes focardii. In general, the manuscript is well-organized with a great interest to the research communities of signaling pheromones and molecular cold-adaptation. However, before it is ready for a publication, I would still recommend several major revisions on results and discussion.

  1. In the activity test, different volumes of chromatographic fractions were added in the culture. Here the concentration of the pheromone in the fractions should be measured.
  2. The authors performed the pheromone activity assay, however, there is no data shown for the activity of the two pheromones from E. focardii. It is recommended to provide the activity testing results and may be even better to present a comparison discussion on the activities of pheromones produced by different Euplotes species.
  3. In the results, comparison analysis was only made on the sequences of the two pheromone gene from E. forcardii. As the authors claimed cold-adaptation in the line 276-286, the gene structures must be compared with those of the genes from other thermo/meso species.
  4. The 3D protein structures are available for several counterparts, it is recommended to perform a protein tertiary structure analyses on the current pheromones, which can be done on the modelling basis and hoped to provide better understanding on the structure adaptation to environmental changes.
  5. In the discussion, the authors mostly focused on the results of the comparison of the different gene structures, however, lacking interpretation of the correlations of evolutionary understanding.

Author Response

(The authors gave the same response as above.)

Round 2

Reviewer 2 Report

The manuscript has significantly improved after revision. Although the protein structure modeling is not available, the rewritten of discussion has provided clear insights in the environmental adaptation through comparison analysis. I would recommend accepting the current manuscript for a publication.